# Hypomyelinated *vps16* Mutant Zebrafish Exhibit Systemic and Neurodevelopmental Pathologies

**DOI:** 10.3390/ijms25137260

**Published:** 2024-07-01

**Authors:** Shreya Banerjee, Shivani Bongu, Sydney P. Hughes, Emma K. Gaboury, Chelsea E. Carver, Xixia Luo, Denise A. Bessert, Ryan Thummel

**Affiliations:** Department of Ophthalmology, Visual and Anatomical Sciences, Wayne State University School of Medicine, Detroit, MI 48201, USA; sbaner15@jhu.edu (S.B.); shivanibongu@wayne.edu (S.B.); sphughes00@gmail.com (S.P.H.); egaboury@alumni.nd.edu (E.K.G.); chelsea.carver@cuanschutz.edu (C.E.C.); xluo@med.wayne.edu (X.L.); dbessert@med.wayne.edu (D.A.B.)

**Keywords:** Vacuolar Protein Sorting 16 (Vps16), zebrafish, mucopolysaccharidosis (MPS), genetic leukoencephalopathy (gLE), behavior, memory, HOPS, CORVET

## Abstract

Homotypic Fusion and Protein Sorting (HOPS) and Class C-core Vacuole/Endosome Tethering (CORVET) complexes regulate the correct fusion of endolysosomal bodies. Mutations in core proteins (VPS11, VPS16, VPS18, and VPS33) have been linked with multiple neurological disorders, including mucopolysaccharidosis (MPS), genetic leukoencephalopathy (gLE), and dystonia. Mutations in human *Vacuolar Protein Sorting 16 (VPS16)* have been associated with MPS and dystonia. In this study, we generated and characterized a zebrafish *vps16(-/-)* mutant line using immunohistochemical and behavioral approaches. The loss of Vps16 function caused multiple systemic defects, hypomyelination, and increased neuronal cell death. Behavioral analysis showed a progressive loss of visuomotor response and reduced motor response and habituation to acoustic/tap stimuli in mutants. Finally, using a novel multiple-round acoustic/tap stimuli test, mutants showed intermediate memory deficits. Together, these data demonstrate that zebrafish *vps16(-/-)* mutants show systemic defects, neurological and motor system pathologies, and cognitive impairment. This is the first study to report behavior abnormalities and memory deficiencies in a zebrafish *vps16(-/-)* mutant line. Finally, we conclude that the deficits observed in *vps16(-/-)* zebrafish mutants do not mimic pathologies associated with dystonia, but more align to abnormalities associated with MPS and gLE.

## 1. Introduction

VPS16 functions in the endolysosomal trafficking system, a dynamic subcellular machinery that depends on endosomal vesicle fusion and fission [1]. VPS16, along with VPS11, VPS18, and VPS33, forms the four-protein component core of two membrane tethering complexes, Homotypic Fusion and Protein Sorting (HOPS) and Class C-core Vacuole/Endosome Tethering (CORVET). These complexes regulate and ensure the fusion of correct endosomal vesicles [2]. Vps16 has been found to be highly conserved from yeast to humans [3], is expressed in multiple cell lines [4], and has a high expression in murine brain tissue [5]. Functionally, Vps16 is specifically responsible for recruiting the SNARE-interacting component Vps33 to the HOPS core complex. This recruitment drives the fusion of late endosomes/autophagosomes with lysosomes [6,7].

Lysosomal storage diseases (LSDs) are inherited, progressive, and multisystemic disorders caused by impaired lysosomal function [8]. Monogenic mutations of lysosomal or non-lysosomal proteins result in an accumulation of lysosomal substrates in various tissues and organs [9]. The clinical manifestations are at infantile stages, with diverse symptoms, and are often associated with physiological and neurological abnormalities [8,10]. Mucopolysaccharidoses (MPSs) are a group of 13 specific LSDs that occur due to defective degradation of mucopolysaccharides or glycosaminoglycans (GAGs), resulting in a build-up of these complex sugar molecules in cells of various tissues [11,12]. Patients typically report coarse facial features, skeletal abnormalities, hepatosplenomegaly, respiratory problems, cognitive impairment, neurological and developmental delays, and excess secretion of urinary GAGs [13]. Successful diagnosis is a long and challenging process, as there is high variability in patient pathology; however, the advent of next-generation sequencing has significantly advanced the identification of causative mutations [8]. 

Towards this end, mutations in the human gene *Vacuolar Protein Sorting 16 (VPS16)* have recently been associated with a novel type of MPS named Mucopolysaccharidosis-Plus Syndrome (MPS-PS) [14]. This is also consistent with previously reported cases of MPS-associated phenotypes in individuals with a missense mutation of *VPS33A* [15,16,17]. Specifically, Sofou et al. 2021 reported two patients from independent families of Iranian and Turkish descent with a homozygous mutation in *VPS16* suffering from MPS [14]. These patients had infantile onset of MPS and presented with progressive psychomotor regression, severe developmental impairment, myelination defects, brain atrophy, skeletal abnormalities, neutropenia, and dysmorphic features [14]. Another report identified two siblings from a consanguineous Turkish–Arabic family suffering from MPS associated with a different homozygous *VPS16* mutation [18]. 

However, several mutations have also linked *VPS16* variants to dystonia-like movement disorders in patients globally [19,20]. Dystonia is a neurological movement disorder characterized by sustained or intermittent muscle contractions resulting in abnormal, repetitive movements and/or postures. Dystonic movements are often initiated or exacerbated by voluntary action and are generally patterned, twisting, and possibly tremulous in nature. Based on clinical characteristics, dystonia can occur as the only symptom or in combination with other neurological or systemic pathology [21,22] which is the case with patients with mutations in *VPS16* [19,20]. 

Complicating the diagnostic odyssey even further is that mutations in *VPS11*—another member of the HOPS/CORVET complex—result in genetic leukoencephalopathy (gLE), a hypomyelination disorder [23]. These individuals show a progressive loss of vision and motor function during the first year of life due to the paucity of myelin in the rapidly developing CNS. Given that all four HOPS/CORVET proteins function in a multiprotein complex and not independently [2,3], the assumption would be that mutations in any of the four would result in similar pathologies; however, this does not appear to be the case [24]. 

Zebrafish is a widely used model to study early neurodevelopmental disorders. Previous work in zebrafish showed that a loss of Vps16 function resulted in hypopigmentation, hypomyelination, and lysosomal defects [14]. A loss of Vps11 function in zebrafish results in identical defects [23,25]. None of the previous studies reported dystonia in these models. Still, the diverse range of pathologies reported in the human cases points to a need for continued characterization of animal models for a loss of HOPS/CORVET function, with a particular focus on determining whether mutant animal models mimic the phenotypes associated with the human pathologies. 

Therefore, the goal of this study was to generate and further characterize a zebrafish *vps16(-/-)* mutant line, with a specific interest in visual and neurological pathologies associated with the aforementioned human cases. We utilized a battery of immunohistochemical and behavioral approaches to test for hypomyelination, CNS cell death, sensorimotor function, and memory. Together, our findings demonstrate that a loss of Vps16 function in zebrafish results in systemic defects, hypomyelination, and neurodevelopmental deficits that align more with MPS and gLE than with dystonia. 

## 2. Results

### 2.1. Zebrafish vps16(-/-) Mutants Exhibit Multiple Systemic Pathologies

We first characterized the gross phenotypes of *vps16(-/-)* mutants during the early stages of development. The initial distinguishing phenotype in the mutants was hypopigmentation, which developed progressively from 3 days post-fertilization (dpf) to 7 dpf (Figure 1A). This is consistent with hypopigmentation patterns in many Vps mutant models, from flies to fish to mice [14,24]. Hypopigmentation was prominent in the retinal pigmented epithelium (RPE) of the eye and in body melanophores of *vps16(-/-)* mutants at 7 dpf (Figure 1A). We also noted that *vps16(-/-)* mutants lacked a developed swim bladder (Figure 1A, asterisk), showed signs of pericardial edema (Figure 1A, arrowhead), and displayed hepatomegaly (Figure 1A, red outline). We next performed survival curves on larval offspring from three separate mating pairs (Figure 1B). Starting at 3 dpf, both hypopigmented mutants and normally pigmented siblings were counted daily for viability. We noted a gradual increase in mortality in the *vps16(-/-)* mutants from 4 to 8 dpf, followed by a dramatic loss of mutants at 11 dpf (Figure 1B). The exact cause of death was unknown. We next examined the CNS for gross histopathology (Figure 1C). Histological sections revealed pyknotic nuclei in the *vps16(-/-)* larval forebrain and retina (Figure 1C, red arrowheads). Furthermore, the mutant retinas showed degeneration of the RPE (Figure 1C, asterisk) and what appeared to be truncated outer segments of photoreceptors (Figure 1C). All of these gross pathologies are consistent with the pathologies reported in *vps11* mutant zebrafish [26]. 

### 2.2. vps16(-/-) Mutants Show Retinal Pathologies by 7 dpf, including Apoptosis, Rod Photoreceptor Truncation, and Müller Glia (MG) Reactive Gliosis

In order to further characterize the retinal pathology of the *vps16(-/-)* mutants, we immunostained 7 dpf retinas with markers for cone and rod photoreceptors, Müller glia (MG), proliferation, and apoptosis. The cone photoreceptor protein Arrestin (Zpr-1) showed a significant upregulation in *vps16(-/-)* mutants (Figure 2A–C), which is consistent with a previous report describing chronic damage of cone photoreceptors [27]. We also noted that rod photoreceptor outer segments (Zpr-3) were significantly truncated in *vps16(-/-)* mutants (Figure 2D–F). Damage to retinal neurons is known to elicit a reactive gliosis response in resident MG, including cellular hypertrophy that is localized to the area of damage [28,29]. Glutamine synthetase (G.S.) immunostaining of MG showed a qualitative and localized expansion of G.S. near damaged photoreceptors; however, quantitative fluorescent intensity measurements did not reach significance (*p* = 0.06) (Figure 2G–I). It is well established that an acute and significant loss of retinal neurons in zebrafish triggers MG to re-enter the cell cycle and produce retinal progenitors [28,30]. However, when we labeled mutant retinas with PCNA, a G1/S cell cycle marker, we noted no evidence of MG proliferation, nor an expansion of the stem cell niche in the ciliary marginal zone (Figure 2J–L). Despite this absence of a regeneration response, TUNEL labeling noted a significantly higher number of apoptotic cells in *vps16(-/-)* mutant retinas compared with their siblings (Figure 2M–O). Together, these data suggest multiple retinal pathologies in *vps16(-/-)* mutant retinas at 7 dpf. 

### 2.3. vps16(-/-) Mutants Exhibit Apoptosis in the CNS and Progressive Hypomyelination

In order to determine whether the pyknotic nuclei we observed in the brains of *vps16(-/-)* mutants were truly apoptotic, we performed TUNEL analysis on sections from *vps16(-/-)* mutant and sibling brains (Figure 3A–I). We observed that mutant brains contained significantly more apoptotic cells in the forebrain (Figure 3A,B), fore-midbrain (Figure 3C,D), mid-hindbrain (Figure 3E,F), and hindbrain (Figure 3G,H) regions. Next, we immunostained *vps16(-/-)* mutant and sibling brains for Myelin Basic Protein (MBP), a marker for compact myelin [31]. We observed no difference in MBP staining in the mid-hindbrain region of siblings and mutants at 5 dpf (Appendix A), but significantly less MBP in mutant brains at 7 dpf (Figure 3J–N), which was especially apparent surrounding the large Mauthner axons in the mid-hindbrain (Figure 3J,K) and hindbrain (Figure 3L,M) regions of a 7 dpf larva. Sofou et al. 2021 also observed hypomyelination in their *vps16* crispant larvae [14]. In addition, progressive hypomyelination was also reported in *vps11* mutant zebrafish [23], which was suggested as a model to study the human hypomyelinating disease genetic leukoencephalopathy [25].

### 2.4. vps16(-/-) Mutants Show Progressive Loss of Sensorimotor Response to Visual Stimuli

In order to determine the impact of the CNS pathologies on larval behavior, we used established behavioral assays to test visuomotor function in *vps16(-/-)* mutant animals at 5 and 7 dpf [25]. The visual stimulation paradigm comprised four alternating cycles of light and dark periods, each lasting 3 min. At 5 dpf, both mutant and sibling larvae showed higher levels of movement in the dark period compared with the light period (Figure 4A,B). In addition, there was no significant difference in the distance traveled (Figure 4C) or the velocity (Figure 4D) of *vps16(-/-)* mutants compared with their siblings. At 7 dpf, both mutants and siblings increased their mobility in light and dark periods (Figure 4E), again with higher levels of movement in the dark (Figure 4F). However, the motility of *vps16(-/-)* mutants at 7 dpf did not increase to the same degree as their siblings (Figure 4E,F). Quantification of movement showed that the mutants traveled a significantly lower distance (Figure 4G) and had lower velocity (Figure 4H) in both light and dark periods when compared with the siblings. These data suggest that the *vps16(-/-)* mutants showed a progressive visuomotor defect between 5 and 7 dpf.

### 2.5. vps16(-/-) Mutants Show a Lower Sensorimotor Response to an Acoustic/Tap Stimulus but Maintain a Habituation Response to a Series of Multiple-Tap Stimuli

To discern between a visuomotor response and a purely motor response, 5 and 7 dpf larvae were subjected to a non-visual acoustic/tap stimulus using the Noldus DanioVision system. At 5 dpf, a low-intensity tap elicited a modest, but discernable increase in mutant larval movement at the tap event (Appendix A, red arrow). In contrast, wild-type siblings showed a robust increase in movement to the tap (Appendix A), which was significantly greater than the distance moved by the mutants (*p* < 0.001; Appendix A). Next, 5 dpf larvae were subjected to a single high-intensity tap. The *vps16(-/-)* mutants showed a greater response than to the low-intensity tap, but the distance moved was still significantly lower than wild-type siblings (*p* < 0.05; Appendix A). Next, we repeated these studies with 7 dpf animals. Again, we found that both wild-type siblings and *vps16(-/-)* mutants responded to both low- (Figure 5A,B) and high-intensity taps (Figure 5C,D), but that siblings showed a significantly greater distanced moved than the mutants (*p* < 0.05). 

In order to understand more complex neurocircuitry in the larvae, we subjected 5 and 7 dpf *vps16(-/-)* mutants and wild-type siblings to a short-term memory habituation paradigm as previously described [25]. This consisted of 10 high-intensity tap stimuli with an ISI of 10 s. Similar to the single-tap stimulus (Figure 5), both *vps16(-/-)* mutants and siblings responded to each of the 10 taps, showing a brief but distinctive increase in distance traveled at every tap, followed by a drop back to baseline movement levels (Appendix A and Figure 6A). Consistent with the single-tap data, we found that the 7 dpf *vps16(-/-)* mutants had a lower overall response compared with the siblings (Figure 6B) and had a significantly lower median distance traveled (Figure 6C). However, both siblings and *vps16(-/-)* larvae showed a habituation response to the 10 consecutive taps (Figure 6D), indicating that *vps16(-/-)* mutants possess a short-term memory recall of at least 10 s. 

### 2.6. vps16(-/-) Mutants Show Abnormalities in Intermediate Memory

Finally, to better understand the intermediate memory of the *vps16(-/-)* mutants at 7 dpf, we adapted a paradigm that was recently shown to effectively test memory in wild-type zebrafish larvae [32]. The stimulation paradigm utilized two rounds of the habituation paradigm described above (Appendix A). “Round 1” contained the identical 10 high-intensity taps that we had utilized previously. Next came an extended period of no stimulation, or a “rest period” of either 2, 5, or 30 min (Appendix A). Finally, “Round 2” of multiple taps was identical to Round 1. After the testing, the habituation curves for each round were statistically compared. A significantly lower habituation curve in the second round indicated that the larvae remembered the previous round of taps during the rest period, and thus had a diminished response to the second round of stimuli. Conversely, statistically identical habituation curves in each round indicated that the larvae forgot the first round of taps during the rest period, and therefore the second round of taps was viewed as a novel stimulus. The results for the 2 min rest period (Figure 7A) showed that wild-type siblings had a significantly lower median distance traveled (*p* < 0.04) and a significantly different habituation curve in Round 2 compared with Round 1 (*p* < 0.0001; Figure 7B). The *vps16(-/-)* mutants did not have significantly different median distance traveled in Round 2 compared with Round 1 but did have a significantly lower habituation curve in Round 2 (*p* < 0.01; Figure 7B). These data suggest that both sibling and mutant larvae remembered the stimulation of Round 1 during the 2 min rest period. 

Next, we increased the rest period to 5 min (Figure 8A). This time, neither group showed a significant difference in the median distance traveled (Figure 8B) and only the sibling larvae exhibited a significantly lower habituation curve in Round 2 compared with Round 1 (Figure 8B). These data suggest that *vps16(-/-)* mutants responded to Round 2 as though it was a novel stimulus. Finally, we extended the rest period to 30 min (Figure 9A). We found that neither sibling nor mutant larvae showed a distinct habituation curve in Round 2 when compared with Round 1 (Figure 9B), suggesting that both groups responded to Round 2 as though it was a novel stimulus. Together, these data suggest that 7 dpf *vps16(-/-)* mutant larvae show distinct differences in their intermediate memory compared to their siblings.

## 3. Discussion

Mutations in the core HOPS/CORVET components and the accessory proteins have been implicated in multiple human pathologies [24], including dystonia [19,20,33], mucopolysaccharidosis (MPS) [14,15,16,17,18], and genetic leukoencephalopathy (gLE) [23,34]. In this study, we characterized a zebrafish *vps16(-/-)* mutant line and found that the loss of Vps16 function resulted in at least some major symptoms from the previously reported diseases [14,20,23]. Here we will highlight the phenotypes of *vps16(-/-)* mutants that are shared with human MPS, as well as similar phenotypes observed among zebrafish *vps11* mutants and patients with gLE.

Sofou et al. 2021 observed multiple systemic defects in their *vps16* crispant model [14]. In our study, we generated a zebrafish *vps16(-/-)* mutant line and confirmed the presence of systemic pathologies in our homozygous mutant larvae, with hypopigmentation being the initial and most prominent gross phenotype. Pigment-producing cells rely heavily on the endolysosomal trafficking pathway, given that melanosomes, the intracellular organelles crucial for pigment development, are derived from early endosomes [35]. Similar to results obtained using the *vps11(plt)* mutant line [26], we found that melanophores were formed in the *vps16(-/-)* mutants; however, they contained a reduced amount of pigment. Pigmentation defects caused by mutations in HOPS/CORVET proteins have been reported in multiple model systems, including *Drosophila* mutant models of *vps16* [36], *vps41* [37], *vps18,* and *vps33* [38,39]. Zebrafish mutation of other HOPS/CORVET core components, namely, *vps18* [40] and *vps11* [26], has produced hypopigmented larvae as well. However, abnormal pigmentation has not been reported as symptoms of gLE, MPS, or dystonia in human patients [14,19,20,23]. This may be due to the different embryonic origins and cellular functions of zebrafish melanophores and human melanocytes [41,42]. Other systemic defects include an underdeveloped swim bladder, hepatomegaly, and pericardial edema (Figure 1A); all these abnormalities were also reported in zebrafish *vps11* and *vps18* mutants [26,40]. The zebrafish swim bladder has a high molecular homology to the mammalian lung [43] and is an established model system to study the hepatobiliary system [44]. Sofou et al. 2021 reported respiratory dysfunction in both patients with MPS studied, but only one had complaints of hepatomegaly and/or splenomegaly; however, no indications of respiratory or hepatic malfunction have been reported in gLE or dystonia [14]. Finally, both *vps11(plt)* and *vps16(-/-)* mutants die by 11 dpf, showing a significantly short life span (Figure 1B) [26]. One of the patients suffering from VPS16-associated MPS died at infantile stages after worsening of pathologies, while others experienced global developmental delays and severely poor quality of life [14,18]. Patients with the *VSP11:C846G* mutation that is causative for gLE live at least into their twenties, although their lifestyle is similarly compromised [23]. It is possible that the very rapid embryonic development of zebrafish does not allow for compensatory mechanisms to overcome the multiple systemic defects, resulting in early death. 

A closer examination of the brain of *vps16* mutants showed a significant increase in neuronal cell death (Figure 3A–H) and hypomyelinated Mauthner axons (Figure 3J–M), which was identical to myelin defects detected in *vps11(plt)* mutants [23]. Cerebral atrophy and myelin defects are major clinical characteristics predominantly observed in MPS and gLE patients [14,23]. In both diseases, patients exhibit hypomyelination and a thin corpus callosum, which are indicative of a slow rate of myelin development [14,23]. The mechanism underlying the hypomyelination phenotype is unknown. However, studies on the human gLE-causing *VPS11:C846G* mutation and MPS-causing *VPS16:N25K* mutation have implicated a decrease in fusion between autophagosomes and lysosomes as a potential mechanism [14,23]. The resultant accumulation of acidic vesicles and the inability to form autolysosomes may cause toxic intracellular conditions, resulting in negative consequences for neurodevelopment.

Larval zebrafish are a powerful tool to study neurodevelopment and behavior. An array of robust swimming and movement patterns are well documented by the age of 7 dpf, which have been used to gain crucial insight into studying mutant lines [45]. A previous study using two *vps11* mutant lines showed progressive sensorimotor deficits similar to gLE patients [25]. Here, we tested whether *vps16(-/-)* mutants exhibited similar behaviors. First, we used an established visual stimulation paradigm of alternating light and dark periods (Figure 4). We found that *vps16(-/-)* mutants showed a progressive loss of normal motor function from 5 to 7 dpf (Figure 4). Although it is currently unclear what definitely caused this phenotype, the significant reduction in myelination of Mauthner axons between 5 and 7 dpf in the mutants (Figure 3N) almost certainly contributed to this defect [46]. Furthermore, substantial cell death in the retina (Figure 2O), the optic tectum, and the hindbrain regions of the brain (Figure 3F,H) could be contributing factors. Notably, visual deficits are primary clinical characteristics in the diagnosis of patients with MPS and gLE but are rare in cases of dystonia [14,20,23]. Therefore, these data are consistent with *vps16(-/-)* mutants showing similar gLE phenotypes as *vps11* mutants and patients with MPS.

Next, we tested the habituation response of *vps16* mutants to multiple high-intensity taps, as we previously demonstrated in *vps11* mutants [25]. It is well established that larval zebrafish elicit a gradual decrease in startle responses to successive acoustic stimuli [47]. Furthermore, the decline in movement between taps is not linear but rather fits to a habituation decay curve [32]. Consistent with this, we observed robust habituation responses from both siblings and *vps16(-/-)* mutants (Figure 6D). We next proceeded to investigate the higher cognitive circuit response of intermediate memory. This is the first report of this assay in any mutant zebrafish line but was based on a similar paradigm using wild-type larvae [32]. Here, 7 dpf larvae were exposed to two “Rounds” of stimulation comprising 10 high-intensity taps with a 10 s ISI with an extended rest period in between (Appendix A). Focus was given to the habituation response to Round 2 of taps following the rest period. It was previously reported that short-term habituation in larval zebrafish is known to last between 25 min and 1 h [48]. However, the duration of habituation is highly dependent on the length and type of the larva training period [47,49,50]. In our assay, the short-term habituation of Round 1 served as the training period and the stimuli in Round 2 served as the test. The 2 min rest period yielded habituation responses for both siblings and mutants that suggested they remembered the first round of stimuli (Figure 7B). It is important to note that the response to the first ‘tap’ event in both Round 1 and Round 2 was very similar as the larvae covered a similar amount of distance (Figure 7B, Figure 8B and Figure 9B). The different habituation response to Round 2 of stimuli is a result of the dramatic drop in movement traveled between the first and second individual ‘tap’ events in Round 2 of taps. This finding is identical to that previously reported [32]. When the rest period was increased to 5 min (Figure 8A), we again found that the sibling larvae remembered the first round of stimuli (Figure 8B). In contrast, there was no difference between the habituation curves in the mutant larval response in Round 1 and Round 2 (Figure 8B), suggesting that the mutants forgot the first set of stimuli in Round 1 and responded to Round 2 as though it were a novel event. Finally, when we extended the rest period to 30 min, neither *vps16(-/-)* nor sibling larvae remembered Round 1 of stimuli (Figure 9B). Therefore, our training paradigm of 10 high-intensity taps with a 10 s ISI was able to define a window of recall in wild-type siblings and differentiate them from the *vps16* mutants. 

Collectively, we show systemic and myelin defects in a zebrafish *vps16(-/-)* model resulting in sensorimotor defects. We also show, for the first time, abnormalities in habituation and memory resulting from a loss of Vps16 function. Diseases associated with *VPS16* mutations present a convoluted picture. While patients with MPS present with myelin and systemic defects, there are no major motor symptoms, which is the primary clinical characteristic of *VPS16* mutations associated with dystonia. Data from our zebrafish *vps16(-/-)* model show a combination of certain pathologies common to all three disorders but are most consistent with phenotypes observed in zebrafish *vps11* mutants, a model for gLE. Future investigations on multiple animal models of Vps mutants are warranted given the complexity of human pathologies associated with their loss of function. As with all animal model studies, the hope is that information gained from these model systems will provide valuable insight into the impact of Vps function on normal development, paving the way for potential interventions.

## 4. Materials and Methods

### 4.1. Zebrafish Lines and Maintenance

Zebrafish *vps16(-/-)* mutants were maintained on an AB background. Fish were fed a combination of brine shrimp and dried flake food three times daily and maintained at 28.5 °C on a 14 h light (250 lux)/10 h dark cycle [51]. All animal care and experimental protocols used in this study were approved by the Institutional Animal Care and Use Committee at Wayne State University School of Medicine (IACUC #21-05-3623). Sex as a biological variable was not considered since zebrafish sex is not determined at the stages of larval development tested in this study.

### 4.2. Generation of the vps16(-/-) Mutant Line 

The target site (5′-GGTGAAGCAGTTGGGATGGA-3′) for the sgRNA was designed to disrupt exon 3 of *vps16*, which was predicted to disrupt the binding of Vps16 to other C-Vps proteins. The sgRNA was prepared by annealing the oligonucleotides (5′-TAGGTGAAGCAGTTGGGATGGA-3′ and 3′-ACTTCGTCAACCCTACCTCAAA-5′) and cloned into *Bsa*I-digested pDR274 (Addgene ID no. 42250). Following sequence confirmation, the plasmid was digested with *Dra*I, and the resultant sgRNA clone was purified. Cas9 nuclease was obtained from pMLM3613 (Addgene ID no. 42251) and linearized with *Pme*I. Both the sgRNA and *Cas9* mRNA were transcribed using mMessage mMachine T7 ultra kit (Cat. no. AM1345, Thermo Fisher Scientific, Waltham, MA, USA). DNaseI treatment and the poly (A) tail reaction were performed according to the manufacturer’s instructions. The sgRNA and the *Cas9*-encoding mRNA were recovered with mini Quick Spin RNA Columns (Cat. no. 11814427001, Roche Diagnostics, Indianapolis, IN, USA). Adult AB fish from among our stocks with 100% identity within the target site and three base pairs (bps) that followed were used for embryo production. Additionally, 1-cell stage embryos were microinjected with a 1:1 solution of the sgRNA and *Cas9* mRNA at a final concentration of 5 ng/µL (sgRNA) and 140 ng/µL (*Cas9* mRNA). Microinjected embryos were raised to 6–8 weeks old, at which point genomic DNA was harvested from individual fin clips and PCR was performed for the target site. A T7 endonuclease I assay was performed to identify heteroduplex PCR products with mismatches. These F0 fish were out-crossed with wild-type AB fish and their F1 offspring were again screened by the T7 endonuclease assay, followed by cloning and sequencing the target site. F1 fish with an identical indel in the target site that contained multiple insertions, deletions, and substitutions were in-crossed to produce the *vps16* mutant line. F2 homozygous fish were sequenced to confirm the *vps16* mutation.

### 4.3. Experimental Design and Statistical Analysis

Experimental approaches were based on previously published and established techniques, with the exception of the intermediate memory paradigm, which we adapted from Beppi et al. 2021 (see below) [32]. Statistical analyses for each approach are clearly defined in the description of the approach. Sample sizes for each data set are detailed in the figure legends. Raw data are available upon request. 

### 4.4. Wholemount Brightfield Imaging

Adult heterozygous *vps16* parents were in-crossed and larvae were raised under standard conditions. The 7 dpf larvae were anesthetized using 1:1000 2-Phenoxyethanol (2-PE) (Cat. no. 77699-250ML, Sigma-Aldrich, St. Louis, MO, USA) and immobilized on glass coverslips using 2% low melting point agarose (Cat. no. V2111, Promega Corporation, Madison, WI, USA) dissolved in 1:1000 2-PE. Coverslips containing embedded larvae were immersed in 1:1000 2-PE solution, and images were collected using a stereo microscope (MZFL III, LeicaMicrosystems, Inc., Buffalo Grove, IL, USA).

### 4.5. Survival Curves

Larvae were separated at 3 dpf into hypopigmented *vps16 (-/-)* mutants and wild-type siblings and were observed daily under a stereo microscope (M165 FC LeicaMicrosystems, Inc., Buffalo Grove, IL, USA) and quantified for the presence of a heartbeat. Survival curve analysis was performed using GraphPad, https://www.graphpad.com/, and statistical differences between the two groups were analyzed using a Mantel–Cox test.

### 4.6. Immunohistochemistry

Immunohistochemistry was performed as described [26] on 14–16 µm frozen brain and eye sections from *vps16(-/-)* mutants and wild-type siblings collected at 7 dpf. Primary antibodies included the following: mouse anti-glutamine synthetase (G.S.) monoclonal antibody (1:500; Chemicon International, Temecula, CA, USA), mouse anti-Zpr-1 monoclonal antibody (1:200; University of Oregon Monoclonal Antibody Facility), mouse anti-Zpr-3 monoclonal antibody (1:200; University of Oregon Monoclonal Antibody Facility), mouse anti-Proliferating Cell Nuclear Antigen (PCNA) monoclonal antibody (1:1000; clone PC10, Sigma-Aldrich, St. Louis, MO, USA), rabbit anti-Myelin Basic Protein (MBP) antisera (1:200; gift from Dr. Bruce Appel, University of Colorado), and mouse anti-HuC/D monoclonal antibody (1:50; clone 16A11, prod. no. A-21271, Thermo Fisher Scientific, St. Louis, MO, USA). AlexaFluor goat anti-rabbit IgG 488 (1:500) and goat anti-mouse IgG 594 (1:500) were used as secondary antisera (Invitrogen-Molecular Probes, Eugene, OR, USA). Nuclei were labeled with TO-PRO-3 (Invitrogen-Molecular Probes, Eugene, OR, USA) at a 1:500 dilution, which was included in the secondary antisera incubation. Tissue sections were mounted with Prolong Gold Antifade Reagent (Cat. no. P10144, Thermo Fisher Scientific, St. Louis, MO, USA). 

### 4.7. Terminal Deoxynucleotidyl Transferase dUTP Nick End Labeling (TUNEL) Analysis

TUNEL was performed using brain and eye tissue sections processed for standard immunohistochemistry. Tissue sections were washed in 1X PBS for 20 min, permeabilized with an ice-cold buffer of 0.1% NaCitrate/0.1% Triton X-100/1X PBS for 2 min and washed in 1X PBS for 5 min at room temperature (RT). Next, sections were incubated in 100 µL of labeling buffer (ApoAlert DNA fragmentation kit; Clontech International, Mountain View, CA, USA) for 10 min at RT, followed by humidified incubation with 50 µL of labeling mix (48 µL labeling buffer, 1 µL of 1 mM biotinylated dNTPs (New England Biolabs, Ipswich, MA, USA) and 1 µL of TdT enzyme (45 U/µL; ApoAlert DNA fragmentation kit; Clontech International, Mountain View, CA, USA), at 37 °C for 1–2 h. The reaction was stopped with a 15 min RT wash with 150 µL of 2X saline-sodium citrate (SSC). Next, tissue sections were washed in 1X PBS and then incubated with Streptavidin conjugated to AlexaFluor 488 (1:200, Invitrogen-Molecular Probes, Eugene, OR, USA) and TO-PRO-3 (1:500, Invitrogen-Molecular Probes, Eugene, OR, USA) diluted with 1X PBS for 1 h in the dark, washed with 1X PBS, and mounted with Prolong Gold Antifade Reagent (Cat. no. P10144, Thermo Fisher Scientific, St. Louis, MO, USA).

### 4.8. Confocal Microscopy and Quantification

For all brain and eye samples in a given experiment, single-plane confocal images were acquired using identical confocal settings on a Leica TCS SP8 confocal microscope (LeicaMicrosystems, Inc., Buffalo Grove, IL, USA). For the quantification of fluorescent pixel density, unaltered confocal images were processed in ImageJ (Ver.no. 154) to isolate the channel containing the fluorescence signal of interest, followed by a calculation of the raw pixel density values within identical ROIs for each analysis group. Pixel intensities were normalized to the wild-type value and graphed in GraphPad. Rod outer segment length was quantified in ImageJ using the measure tool that was previously calibrated to a micron scale bar. Finally, hand counts of TUNEL-positive cells were performed in both the brain and the retina. Significant differences between groups were determined using a Student’s *t*-test in GraphPad with a *p* < 0.05 used as a cutoff for significance. 

### 4.9. Histopathology

Eyes from euthanized fish were fixed with 2.5% glutaraldehyde (Ted Pella, Inc., Redding, CA, USA) in a 1:1:1 mixture of 0.1 M phosphate buffer (pH 7.4), 0.2 M phosphate buffer, and water overnight at 4 °C. Samples were subsequently transferred to a fixative containing 3% osmium tetroxide (Electron Microscopy Sciences, Hatfield, PA, USA), 2.5% glutaraldehyde, and 0.2 M phosphate buffer in a 1:1:1 ratio on ice twice for 1.5 hrs. Samples were rinsed with 0.1 M phosphate buffer twice for 15 min, and then dehydrated in increasing gradients of ethanol followed by propylene oxide (Sigma-Aldrich, St. Louis, MO, USA). Samples were infiltrated and embedded in Epon-Araldite (Electron Microscopy Sciences, Hatfield, PA, USA). Sections (1.0 µm) were cut on a Leica EM UC6 microtome (LeicaMicrosystems, Inc., Buffalo Grove, IL, USA) and stained with a 1:1 mixture of 1% Toluidine Blue (Electron Microscopy Sciences) and 1% Pyronin B (Hopkins & Williams Chadwell Health, Essex, England) at 43 °C. Sections were imaged with a Leica DM4000B microscope (LeicaMicrosystems, Inc., Buffalo Grove, IL, USA) and each image was adjusted for brightness/contrast in Adobe Photoshop using identical levels of adjustment.

### 4.10. Behavioral Analysis

A behavioral analysis of *vps16(-/-)* larvae and wild-type siblings was performed at 5 dpf and 7 dpf as previously described [25]. Briefly, we used the DanioVision Observation Chamber (Noldus Information Technology, Wageningen, The Netherlands) linked with the EthoVision XT13 software, and larval movements were tracked using a Basler Gen1 Camera (Basler acA1300-60, Exton, PA, USA). At 4 or 6 dpf, individual larvae (12 mutants and 12 siblings) were placed in clear 24-well plates (diameter 1.65 cm wells) to be acclimatized in the well plates for 24 h prior to behavioral analysis on the following day. All recordings were performed the next day between 1 and 4 PM. The stimulus paradigms used for this study are described below.

#### 4.10.1. Light/Dark Paradigm

Placed inside the DanioVision Observation Chamber, larvae were subjected to an initial 12 min acclimation period in the dark, followed by four alternating periods of 3 min light (10,500 lux, illuminated uniformly from below the plates) and 3 min dark. The response to this stimulation was recorded as distance moved (cm) and velocity (cm/s) during each 3 min stimulus period. Heat maps representing the total movement of individual larvae during the 3 min stimulus periods were also collected. The average distance and average velocity were calculated using Microsoft Excel, https://www.microsoft.com/en-ie/microsoft-365/excel, and GraphPad and statistical differences between the mutants and siblings during the first light and dark period were determined using a Mann–Whitney test with a *p* < 0.05 as a cutoff for significance.

#### 4.10.2. Single Acoustic/Tap Paradigm

Two pre-set tap settings in the DanioVision Observation Chamber were used—intensity 3 (low-intensity tap) or intensity 8 (high-intensity tap). The larvae were subjected to an initial 12 min acclimation period, followed by 1 min of no stimulation, a single tap at low or high intensity, and finally an additional 1 min of no stimulation. The response to stimulation was recorded as distance moved (cm) for individual larvae. The average distance traveled at the tap event was calculated using Microsoft Excel and GraphPad, and statistical differences between the mutants and siblings were determined using a Mann–Whitney test with a *p* < 0.05 as a cutoff for significance.

#### 4.10.3. Multiple Acoustic/Tap Habituation Paradigm

The larvae were subjected to an initial 12 min acclimation period, followed by 1 min of no stimulation, then 10 consecutive high-intensity taps at 10 s interstimulus intervals (ISIs) each, and finally, an additional 1 min of no stimulation. The response to stimulation was recorded as distance moved (cm) for individual larvae. Habituation curves were calculated using GraphPad and statistical differences within and between the groups were determined via a one-phase exponential decay as previously described [25]. In addition, the distance traveled during the series of taps was calculated using Microsoft Excel and GraphPad, and statistical differences between the mutants and siblings were determined using a Welch’s *t*-test with a *p* < 0.05 as a cutoff for significance.

#### 4.10.4. Intermediate Memory Paradigm

The larvae were subjected to an initial 12 min acclimation period, followed by “Round 1” of 10 consecutive high-intensity taps administered at an ISI of 10 s (Appendix A). This was followed by an extended period of no stimulation, termed as “rest period”. Rest periods of either 2, 5, or 30 min were used. Finally, “Round 2” stimulation of 10 high-intensity taps was administered at the identical ISI of 10 s. Round 1 and Round 2 stimulation were identical with respect to tap intensity and ISIs. The response to stimulation was recorded as distance moved (cm) for individual larvae. Habituation curves were calculated using GraphPad. Statistical differences between the response to Round 1 and Round 2 stimuli were determined via a one-phase exponential decay as previously described [32]. In addition, the median distance traveled during the series of taps was calculated using Microsoft Excel and GraphPad, and statistical differences between Round 1 and Round 2 were determined using a Welch’s *t*-test with a *p* < 0.05 as a cutoff for significance. As described in Beppi et al. 2021, if the behavioral response to Round 2 is significantly different than that of Round 1, then the larva retained a memory of the stimuli from Round 1 during the rest period [32]. 

## Figures and Tables

**Figure 1 ijms-25-07260-f001:**
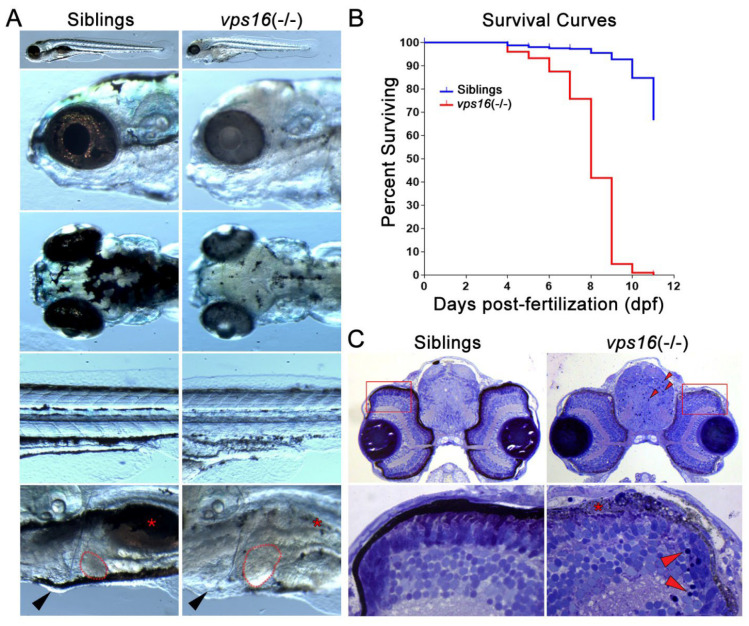
*vps16(-/-)* mutants can be phenotypically distinguished from wild-type siblings. (**A**) Images of wild-type sibling (**left**) and *vps16(-/-)* (**right**) larvae displaying distinctive pigmentation patterns at 7 dpf. Sibling larva (**left**) with characteristic darkly pigmented melanophore in RPE layer within eye and darkly pigmented melanophore aggregates on top of head and near yolk sac extension on side of body. Location of normally pigmented swim bladder (red asterisk), size of liver (outlined in red), and absence of edema (black arrowhead) shown in sibling larvae. *vps16(-/-)* mutant larva (**right**) showing severe hypopigmentation in RPE and reduced pigmentation on head and near yolk sac extension. *vps16(-/-)* mutant displaying hypopigmented and reduced swim bladder (red asterisk), hepatomegaly (outlined in red), and pericardial edema (black arrowhead). (**B**) Survival curve of siblings (blue) and *vps16(-/-)* mutants (red) depicting death of all mutants by 11 dpf. (**C**) Histological sections of 7 dpf sibling and *vps16(-/-)* larval forebrain near optic nerve showing pyknotic nuclei in mutant brain and retina (red arrowheads). *vps16(-/-)* larval retina shows degeneration and absence of pigmented RPE and truncated photoreceptors (asterisks) and lack of distinct organization of photoreceptor outer segments.

**Figure 2 ijms-25-07260-f002:**
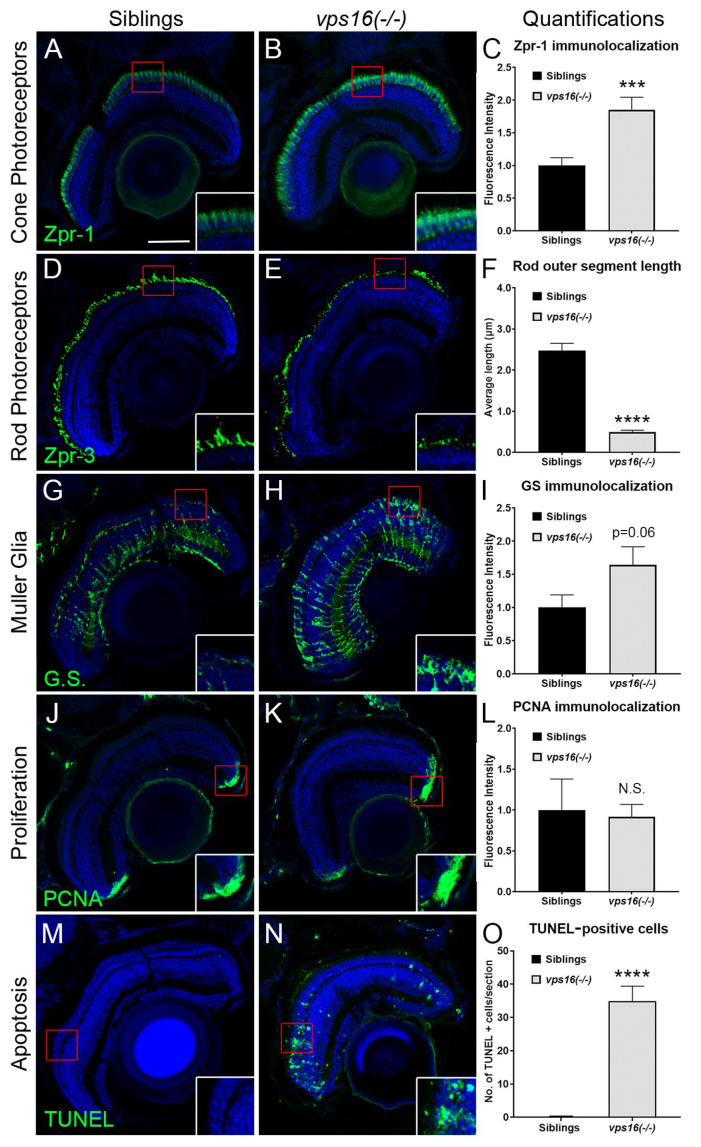
*vps16(-/-)* mutants at 7 dpf show retinal pathology including photoceptor degeneration, reactive gliosis of Müller glia (MG), and cell death. (**A**,**B**) Immunolocalization of Arrestin (Zpr-1) (green) in double cone photoreceptors. (**A**) Siblings (n = 15) show normal Zpr-1 immunolabeling in perinuclear domain of red/green double cones. (**B**) Zpr-1 immunolabeling observed throughout cytoplasm of hypertrophied and degenerating cones in *vps16(-/-)* mutants (n = 18). (**C**) Graph representing quantification of Zpr-1 immunolocalization in 7 dpf siblings and *vps16(-/-)* mutants. *** *p* = 0.0009. (**D**,**E**) Immunolocalization of Zpr-3 (green) in outer segments of rod photoreceptors. (**D**) Outer segments of healthy rod photoreceptors in siblings (n = 13) show normal length and structure. (**E**) Truncated outer segments of degenerating rod photoreceptors observed in *vps16(-/-)* mutants (n = 11). (**F**) Graph representing quantification of average length of rod outer segments immunolabeled with Zpr-3 in 7 dpf siblings and *vps16(-/-)* mutants. **** *p* < 0.0001. (**G**,**H**) Immunolocalization of glutamine synthetase (G.S.) (green) in MG. (**G**) Normal levels of G.S. immunolabeling in outer regions of MG of siblings (n = 14). (**H**) G.S. immunolabeling in ends of MG in outer retina showing reactive gliosis in *vps16(-/-)* mutants (n = 18). (**I**) Graph representing quantification of G.S. immunolocalization in 7 dpf siblings and *vps16(-/-)* mutants. *p* = 0.06. (**J**,**K**) Immunolocalization of PCNA-positive retinal progenitors (green) in CMZ. (**J**) PCNA-positive retinal progenitors shown in CMZ of siblings (n = 9). (**K**) Normal population of PCNA-positive retinal progenitors shown in CMZ of *vps16(-/-)* mutants (n = 8). (**L**) Graph representing quantification of PCNA immunolocalization in retinal CMZ of 7 dpf siblings and *vps16(-/-)* mutants. “N.S.” = not significant. (**M**–**O**) TUNEL-positive cells (green) in retinal sections. (**M**) Minimal to no TUNEL-positive apoptotic cells seen in sibling retina (n = 16). (**N**) Large number of TUNEL-positive apoptotic cells observed in *vps16(-/-)* mutant retina (n = 16), specifically in outer nuclear layer. (**O**) Graph representing quantification of average number of TUNEL-positive apoptotic cells in retinal sections of 7 dpf siblings and *vps16(-/-)* mutants. **** *p* < 0.0001. Scale bar in panel A = 50 microns for all images. Blue = TO-PRO-3, a nuclear stain. Error bars indicate SEM.

**Figure 3 ijms-25-07260-f003:**
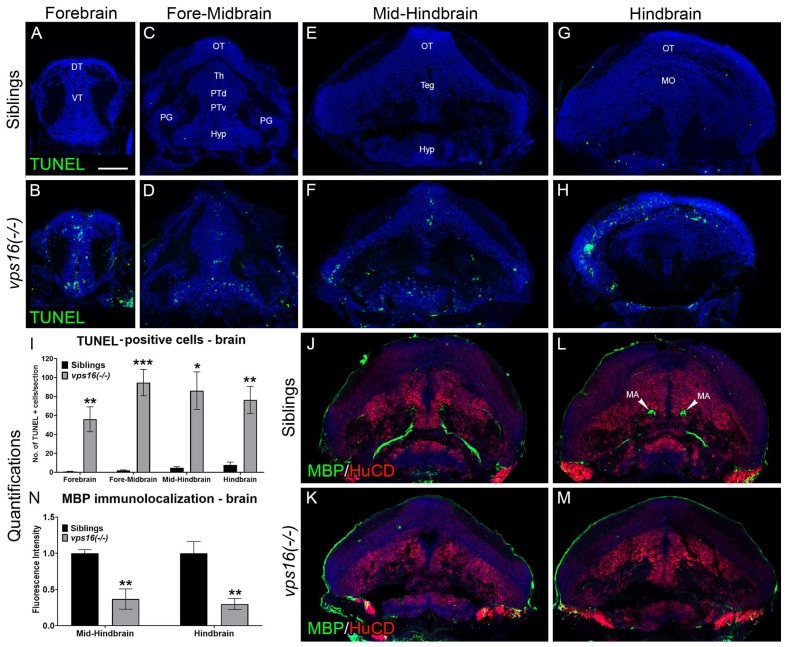
*vps16(-/-)* mutants show significant cell death and hypomyelination by 7 dpf. (**A**–**I**) TUNEL assay performed on brain sections of 7 dpf sibling and *vps16(-/-)* mutant larvae. Brain sections identified and sub-categorized in four regions: (**A**,**B**) forebrain, (**C**,**D**) fore-midbrain, (**E**,**F**) mid-hindbrain, and (**G**,**H**) hindbrain. (**A**,**C**,**E**,**G**) Minimal to no TUNEL-positive apoptotic cells (green) seen in sibling brain regions (n: 10 forebrain, 7 fore-midbrain, 9 mid-hindbrain, 4 hindbrain). (**B**,**D**,**F**,**H**) Numerous TUNEL-positive apoptotic cells (green) seen in *vps16(-/-)* mutant brain regions (n = 10 forebrain, *p* = 0.0022; n = 7 fore-midbrain, *p* = 0.0005; n = 5 mid-hindbrain, *p* = 0.0146; n = 7 hindbrain, *p* = 0.0027). (**I**) Graph representing quantification of average number of TUNEL-positive apoptotic cells in specific brain regions of 7 dpf siblings and *vps16(-/-)* mutants. (**J**–**N**) Immunolocalization of MBP (myelin, green) and HuCD (neurons, red) in brain section from two regions of sibling and *vps16(-/-)* mutant larvae: (**J**,**K**) mid-hindbrain and (**L**,**M**) hindbrain. (**J**–**L**) MBP immunolocalizes to Mauthner axons in mid-hindbrain (n = 6) and hindbrain (n = 6) regions of sibling brain. (**K**,**M**) MBP immunolocalization in mid-hindbrain (n = 6, *p* = 0.0048) and hindbrain (n = 6, *p* = 0.0061) regions of *vps16(-/-)* mutant brain showing decrease in myelin content. (**N**) Graph representing quantification of MBP immunolocalization in specific brain regions of 7 dpf siblings and *vps16(-/-)* mutants. Scale bar in panel A = 50 microns for all images. Blue = TO-PRO-3, nuclear stain. Error bars indicate SEM. * *p* < 0.05, ** *p* < 0.01, *** *p* < 0.001.

**Figure 4 ijms-25-07260-f004:**
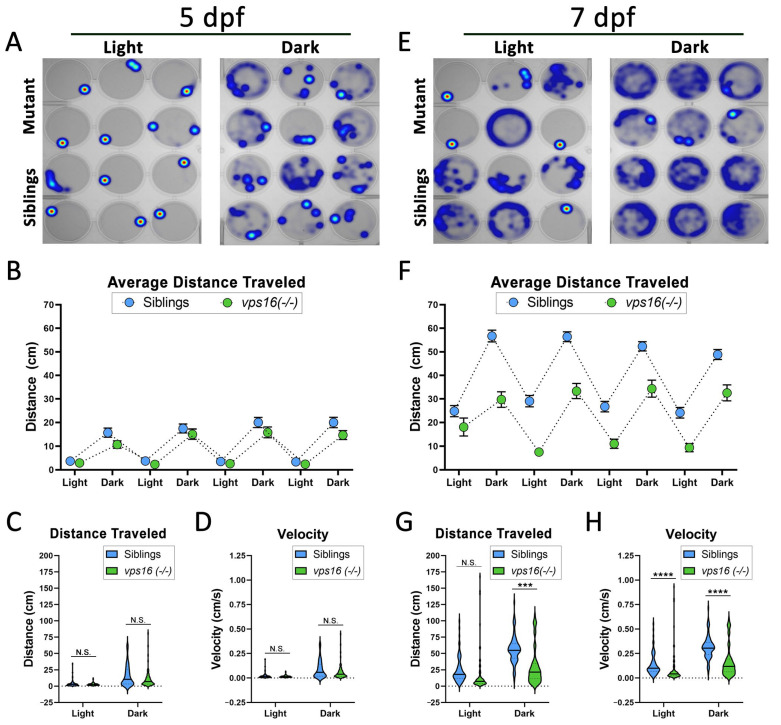
*vps16(-/-)* mutants show normal distance traveled at 5 dpf and significantly reduced distance traveled at 7 dpf in response to alternating cycles of light and dark. (**A**) Representative heatmap images of total distance moved by individual *vps16(-/-)* larvae at 5 dpf (mutants = top two rows, siblings = bottom two rows) during 3 min period of light and dark. (**B**) Graph representing average distance traveled by *vps16(-/-)* larvae at 5 dpf in 4 alternating light/dark cycles of 3 min each (siblings in blue; mutants in green). (**C**) Graph representing average distance traveled by siblings (blue, n = 71) and *vps16(-/-)* (green, n = 72) in light and dark periods. (**D**) Graph representing average velocity of siblings (blue, n = 71) and *vps16(-/-)* (green, n = 72) in light and dark periods. (**E**) Representative heatmap images of total distance moved by individual *vps16(-/-)* larvae at 7 dpf (mutants = top two rows; siblings = bottom two rows) during 3 min period of light and dark. (**F**) Graph representing average distance traveled by *vps16(-/-)* larvae at 7 dpf in 4 alternating light/dark cycles of 3 min each (siblings in blue; mutants in green). (**G**) Graph representing average distance traveled by siblings (blue, n = 72) and *vps16(-/-)* (green, n = 72) in light and dark periods. (**H**) Graph representing average velocity of siblings (blue, n = 72) and *vps16(-/-)* (green, n = 72) in light and dark periods. For all graphs, “N.S.” = not significant, “***” = *p* < 0.001, “****” = *p* < 0.0001, and error bars indicate SEM.

**Figure 5 ijms-25-07260-f005:**
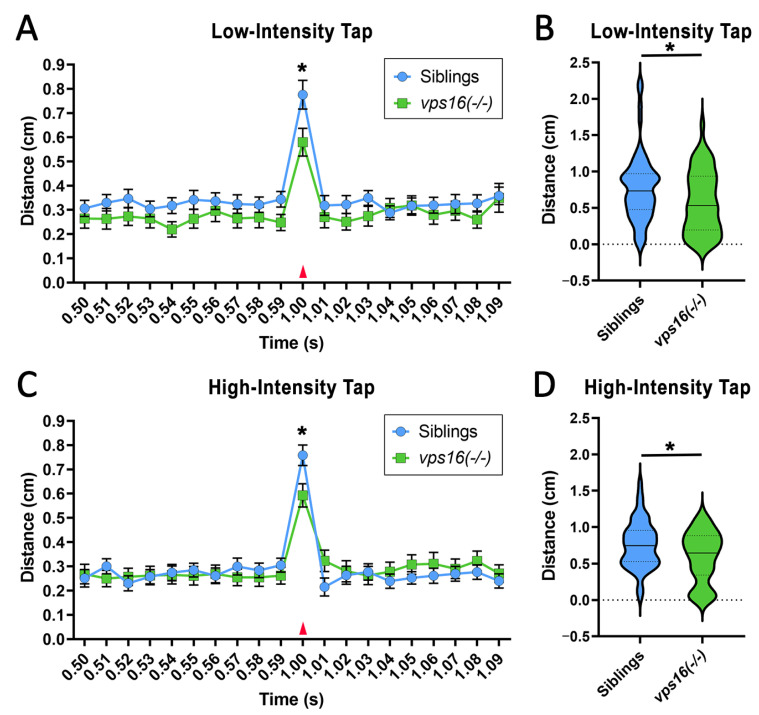
*vps16(-/-)* mutants at 7 dpf show significantly reduced distance traveled in response to single acoustic/tap stimulus at both low and high intensity. (**A**) Line graph representing average distance traveled by *vps16(-/-)* larvae (green, n = 56) and siblings (blue, n = 60) at 7 dpf in response to single low-intensity acoustic/tap stimulus (red arrow). (**B**) Violin plots representing average distance traveled by *vps16(-/-)* larvae (green, n = 56) and siblings (blue, n = 60) at 7 dpf in response to single low-intensity acoustic/tap stimulus. *p* < 0.05. (**C**) Line graph representing average distance traveled by *vps16(-/-)* larvae (green, n = 53) and siblings (blue, n = 58) at 7 dpf in response to single high-intensity acoustic/tap stimulus (red arrow). (**D**) Violin plots representing average distance traveled by *vps16(-/-)* larvae (green, n = 53) and siblings (blue, n = 58) at 7 dpf in response to single high-intensity acoustic/tap stimulus. For all graphs, “*” = *p* < 0.05 and error bars indicate SEM.

**Figure 6 ijms-25-07260-f006:**
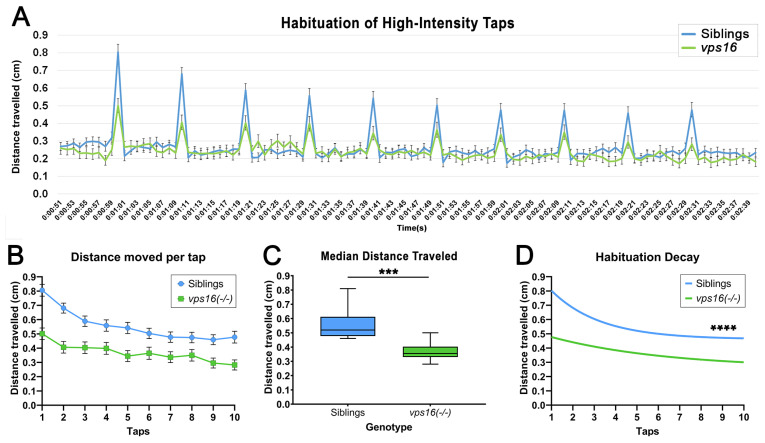
*vps16(-/-)* mutants at 7 dpf show significantly reduced habituation response to multiple high-intensity acoustic/tap stimuli compared with their siblings. (**A**) Line graph representing distance traveled by *vps16 (-/-)* larvae at 7 dpf during complete high-intensity multiple-tap paradigm comprising 10 high-intensity taps with 10 s interstimulus intervals (ISIs) (sibling in blue, *vps16(-/-)* in green). (**B**) Graph representing average distance moved by 7 dpf siblings (blue; n = 84) and *vps16(-/-)* (green; n = 78) larvae at each individual tap stimuli. (**C**) Box plots representing median distance traveled by 7 dpf siblings (blue; n = 84) and *vps16(-/-)* (green; n = 78) larvae in response to multiple-tap stimuli. “***” = *p* < 0.001. (**D**) First-order exponential decay curve representing average distance traveled by 7 dpf siblings (blue; n = 84) and *vps16(-/-)* (green; n = 78) larvae at each individual tap stimuli. “****” = *p* < 0.0001. Error bars indicate SEM.

**Figure 7 ijms-25-07260-f007:**
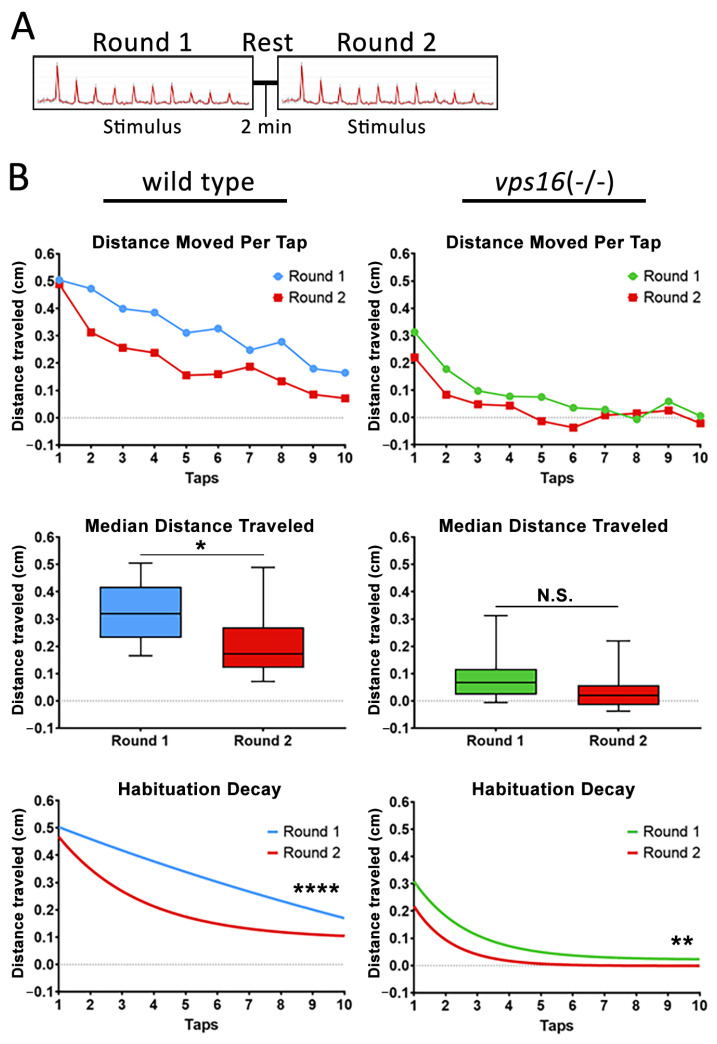
Both *vps16(-/-)* mutants and sibling larvae at 7 dpf show distinct habituation response post 2 min rest. (**A**) Schematic representation of intermediate memory paradigm showing round one stimulus having 10 high-intensity taps, followed by 2 min rest period with no stimulation, and finally round two stimulus having 10 high-intensity taps. (**B**) Line graph representing average distance traveled in response to each individual tap by 7 dpf siblings (blue, n = 105) and *vps16(-/-)* mutants (green, n = 107) before and after 2 min rest period (red). Box plots representing median distance traveled by 7 dpf siblings (blue, n = 105) and *vps16(-/-)* (green, n = 107) larvae in response to multiple-tap stimuli before and after 2 min rest period (red). First-order exponential decay curve representing average distance traveled in response to round one multiple-tap stimuli by 7 dpf siblings (blue, n = 105) and *vps16(-/-)* mutant (green, n = 107) larvae before and after 2 min rest period (red). For all graphs, “N.S.” = not significant, “*” = *p* < 0.05, “**” = *p* < 0.01, “****” = *p* < 0.0001, and error bars indicate SEM.

**Figure 8 ijms-25-07260-f008:**
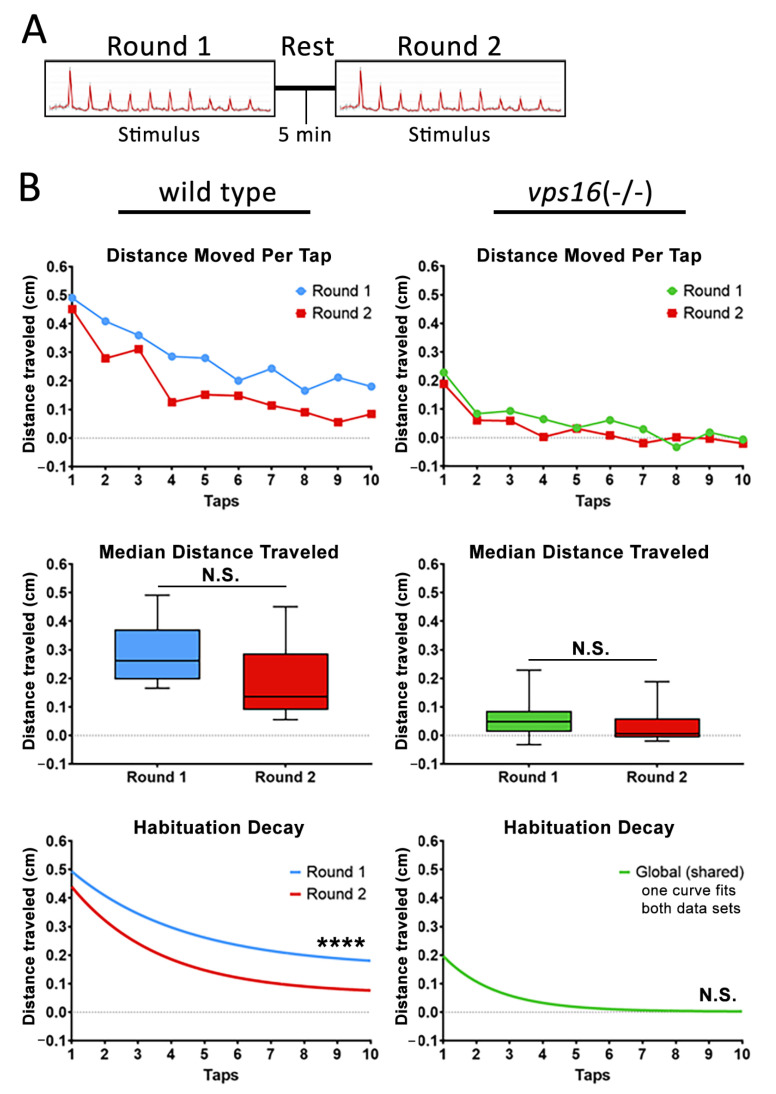
Sibling larvae at 7 dpf do show distinct habituation response post 5 min rest, but *vps16(-/-)* mutants do not show distinct response. (**A**) Schematic representation of intermediate memory paradigm showing round one stimulus having 10 high-intensity taps, followed by 5 min rest period with no stimulation, and finally round two stimulus having 10 high-intensity taps. (**B**) Line graph representing average distance traveled in response to each individual tap by 7 dpf siblings (blue, n = 60) and *vps16(-/-)* mutants (green, n = 59) larvae before and after 5 min rest period (red). Box plots representing median distance traveled by 7 dpf siblings (blue, n = 60) and *vps16(-/-)* (green, n = 59) larvae in response to multiple-tap stimuli before and after 5 min rest period (red). First-order exponential decay curve representing average distance traveled in response to round one multiple-tap stimuli by 7 dpf siblings (blue, n = 60) and *vps16(-/-)* mutant (green, n = 59) larvae before and after 5 min rest period (red). For all graphs, “N.S.” = not significant, “****” = *p* < 0.0001, and error bars indicate SEM.

**Figure 9 ijms-25-07260-f009:**
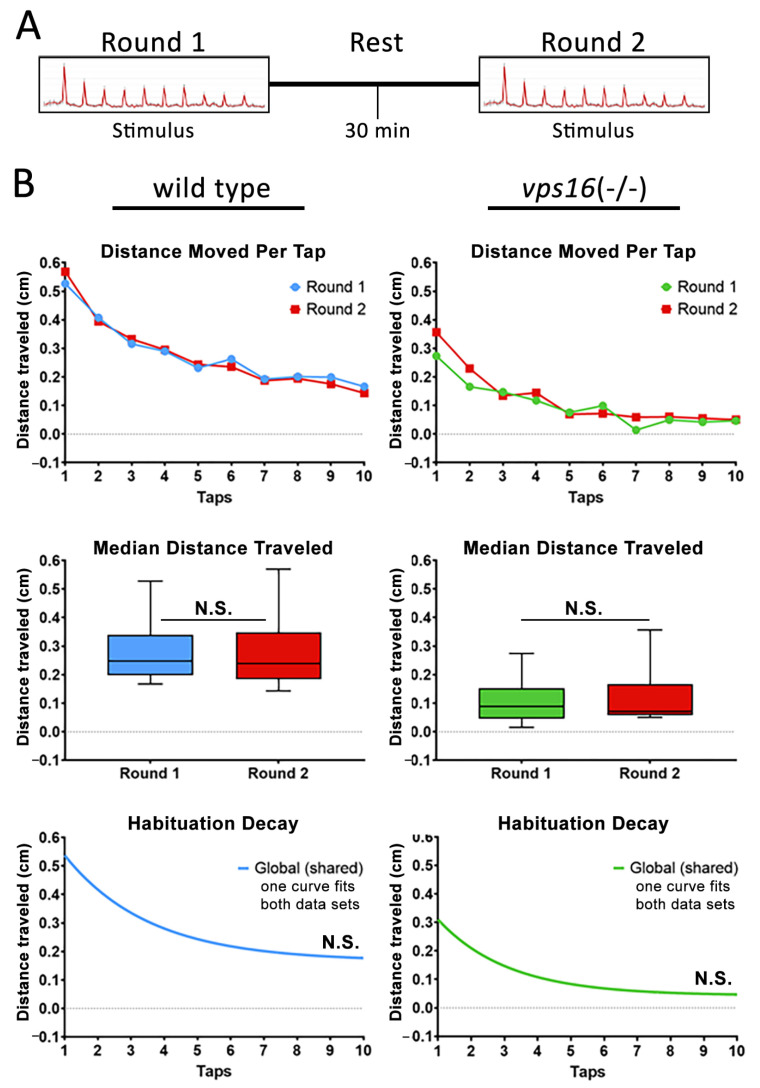
Both sibling larvae and *vps16(-/-)* mutants at 7 dpf do not show distinct habituation response post 30 min rest. (**A**) Schematic representation of intermediate memory paradigm showing round one stimulus having 10 high-intensity taps, followed by 30 min rest period with no stimulation, and finally round two stimulus having 10 high-intensity taps. (**B**) Line graph representing average distance traveled in response to each individual tap by 7 dpf siblings (blue, n = 72) and *vps16(-/-)* mutants (green, n = 68) before and after 30 min rest period (red). Box plots representing median distance traveled by 7 dpf siblings (blue, n = 72) and *vps16(-/-)* (green, n = 68) larvae in response to multiple-tap stimuli before and after 30 min rest period (red). First-order exponential decay curve representing average distance traveled in response to round one multiple-tap stimuli by 7 dpf siblings (blue, n = 72) and *vps16(-/-)* mutant (green, n = 68) larvae before and after 30 min rest period. For all graphs, “N.S.” = not significant, and error bars indicate SEM.

## Data Availability

Raw data supporting the findings of this study will be made available on a reasonable request sent to the corresponding author.

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
