# Peer review of "Hypomyelinated vps16 Mutant Zebrafish Exhibit Systemic and Neurodevelopmental Pathologies"

_ijms, 2024, doi:10.3390/ijms25137260_

Round 1
Reviewer 1 Report
Comments and Suggestions for Authors
Reviewer #1:The paper by Shreya et al., describing the relationship between vps16 and neurodevelopmental pathologies, is very good. The mechanism of vsp16 mutation leading to multiple organ developmental arrest and reduced ability to respond to external environmental stimuli in zebrafish was briefly described. The apoptosis of brain and periocular cells determined by TUNEL staining may be closely related to the stimulation induction of behavioral analysis. Overall the introduction is clear, and the flow and logic are good.
My biggest criticism of the manuscript is that deficiency of VPS16 resulted in multiple phenotypes. Specific to brain and eye development, vsp16 deficiency resulted in a strong inhibition of cells and a large increase in apoptosis. I suggest that the characteristic Maker of different cells should be performed by q-RT PCR to determine which cell development vsp16 mainly affects. I suggest that in situ hybridization should also be used to explore the relationship between VSP16 expression patterns and brain and eye development.
Comments on the Quality of English LanguageThe English introduction was fluent, the logic was clear, and no major grammatical and word order errors were found.
Author Response
Thank you for your comments. Given the timeframe, we focused our efforts on the items that were marked “Must be improved” by the reviewers. You did not have any of these, but we wanted to respond to your comments to be respectful.
Comment 1. Overall the introduction is clear, and the flow and logic are good.
Response 1. Thank you. But please note that both of the other reviewers requested that we expand the introduction. We have done so in the revised manuscript, if you want to take a look
Comment 2. My biggest criticism of the manuscript is that deficiency of VPS16 resulted in multiple phenotypes. Specific to brain and eye development, vsp16 deficiency resulted in a strong inhibition of cells and a large increase in apoptosis. I suggest that the characteristic Maker of different cells should be performed by q-RT PCR to determine which cell development vsp16 mainly affects. I suggest that in situ hybridization should also be used to explore the relationship between VSP16 expression patterns and brain and eye development.
Response 2. Indeed, both the fish and the human cases results in multiple phenotypes/pathologies. In our revised introduction we have clarified this for the reader. In regards to doing qPCR or in situ, we felt that the immunohistochemistry did a better job of showing the cell types than these two assays.
Reviewer 2 Report
Comments and Suggestions for Authors
This paper constructed a vsp6-/- zebrafish model and observed its behavior and morphology changes. The data shows that loss of vsp6 enhances apoptosis and impairs brain functions and related behaviors. The major concern is the loss of mechanisms and the novelty of neurodevelopmental pathologies. 1.The zebrafish starts to die at 3 dpf, and neuronal phenotypes are observed at 7 dpf. Vsp16 is important for muscle development, and pericardial edema was observed, suggesting a potential effect on the heartbeat. If there are abnormalities in the heartbeat, it could lead to zebrafish death, and apoptotic activity may be enhanced in various tissues (such as the retina and neurons). So when does the heartbeat change? And how about apoptosis in other tissues, like hematopoiesis, at day 7? 2. Sofou et al., 2021, also demonstrated that the loss of VPS16 causes impaired myelination and accumulation of autophagosomes in zebrafish at 3 dpf. Does this consistent with your results? Additionally, does the accumulation of autophagosomes contribute to the enhanced apoptosis observed in your study?
Comments on the Quality of English LanguageOverall looks good.
Author Response
Thank you for your comments. Given the timeframe, we focused our efforts on the items that were marked “Must be improved” by the reviewers. You indicated revisions should be done to the Introduction and Results.
Comment 1. If there are abnormalities in the heartbeat, it could lead to zebrafish death, and apoptotic activity may be enhanced in various tissues (such as the retina and neurons). So when does the heartbeat change? And how about apoptosis in other tissues, like hematopoiesis, at day 7?
Response 1. We felt that a thorough analysis of hematopoiesis and heartbeat rates was beyond the scope of this work. But we do not note any heartbeat differences at 5 and 7 dpf, when the behavioral assays were completed. We due note a change in heartbeat beyond 8 dpf, so it is possible that the reviewer is correct and that this is associated with cause of death.
Comment 2. Sofou et al., 2021, also demonstrated that the loss of VPS16 causes impaired myelination and accumulation of autophagosomes in zebrafish at 3 dpf. Does this consistent with your results?
Response 2. To satisfying the reviewers suggestion, we performed additional MBP staining on 5 dpf embryos. These data are found in a new Supplemental Figure 1 and in lines 191-195, where we also reference the Sofou work. We found no difference in MBP expression at 5 dpf, but like Sofou et al, significant hypomyelination at 7dpf in the mutants (Figure 3). This progressive hypomyelination was also observed in vps11 mutants.
Comment 3. Additionally, does the accumulation of autophagosomes contribute to the enhanced apoptosis observed in your study?
Response 3. We don’t know. WB analysis could be done with protein ground up from whole larvae, but in order to localize autophagosomes to a tissue of interest, we would need to mate the Tg(Lc3-EGFP) line onto our mutant background. That would take 6 months, which is beyond the scope of this work. We did accomplish this in our vps11 mutant and found accumulation of GFP+ autophagosomes in the RPE of the retina.
Reviewer 3 Report
Comments and Suggestions for Authors
Very detailed paper with lots of experiments using an animal model.
The limitation of the paper is that the results obtained in zebrafish are not entirely reproducible in humans.
The reader finds it difficult to report the usefulness of the results obtained on zebrafish for usefulness to humans, even if this is well reflected in the conclusions.
I have no observations on the results, but I ask the authors to broaden the introduction by dividing it into two: pathologies in zebrafish and in humans. With these explanatory modifications the reader will be able to understand what role knock out of the gene has in the chosen animal model and the repercussions on humans.
Author Response
Thank you for your comments. Given the timeframe, we focused our efforts on the items that were marked “Must be improved” by the reviewers. You indicated revisions should be done to the Introduction.
Comment 1. I ask the authors to broaden the introduction by dividing it into two: pathologies in zebrafish and in humans.
Response 1. We have done as you requested.
Round 2
Reviewer 2 Report
Comments and Suggestions for Authors
The authors have addressed the reviewers' primary concerns and incorporated additional data that enhances the manuscript's overall quality. While the study may not introduce groundbreaking novelty, it presents advancements and thorough analyses that merit publication.
Comments on the Quality of English LanguageThe manuscript is written in clear and understandable English.